# Can Strict Protection Stop the Decline of Mangrove Ecosystems in China? From Rapid Destruction to Rampant Degradation

**Wenqing Wang [1], Haifeng Fu [1], Shing Yip Lee [2], Hangqing Fan [3] and Mao Wang [1,*]**

[1]   Key Laboratory of the Coastal and Wetland Ecosystems, Ministry of Education,
    College of the Environment & Ecology, Xiamen University, Xiamen 361102, China;
    mangroves@xmu.edu.cn (W.W.); peakandsea@163.com (H.F.)
[2]   School of Life Sciences, The Chinese University of Hong Kong, Shatin, Hong Kong, China;
    joesylee@cuhk.edu.hk
[3]   Guangxi Mangrove Research Center, Beihai 536000, China; fanhq666@126.com
[*]   Correspondence: wangmao@xmu.edu.cn; Tel.: +86-136-6603-7893

**Abstract:** China has lost about 50% of its mangrove forests from 1950 to 2001. Since 2001, mangrove forest area has increased by 1.8% per year due to strict protection of the remaining mangrove forests and large-scale restoration. By 2019, 67% of the mangrove forests in China had been enclosed within protected areas (PAs). In terms of the proportion of PAs of mangrove forests, China has achieved the conservation target of "Nature Needs Half". The ongoing degradation of mangrove forests was assessed at the species, population, community and ecosystem levels. The results show that despite the strict protection, the remaining mangrove forests are suffering extensive degradation due to widespread anthropogenic disturbance. Of the 26 mangrove species, 50% are threatened with extinction, a proportion higher than the average for all higher plants in China (10.8%). Local extinction of some common species like *Bruguiera gymnorhiza* is widespread. About 53% of the existing mangrove areas were dominated by low-intertidal pioneer species. Consequently, the carbon stock in vegetation has decreased by 53.1%, from 21.8 Tg C in the 1950s to 10.2 Tg C in 2019. Meanwhile, there is an estimated 10.8% concomitant decrease in the carbon sequestration rate. The root cause for this degradation in China is seawall construction because most mangroves are outside seawalls in China. Without fundamental changes in protection and restoration strategies, mangrove forests in China will continue to degrade in spite of strict protection and large-scale restoration. Future mangrove conservation effort should aim to preserve the diversity of both the biota and the ecological processes sustaining the mangrove ecosystem. A few suggestions to raise the effectiveness of mangrove conservation actions were provided.

**Keywords:** mangrove forest; conservation; restoration; biodiversity; seawall; protection

## 1. Introduction

Biodiversity loss and ecosystem degradation are the two major environmental problems resulting from anthropogenic disturbances and global climate change [1–4]. Major conservation efforts to reverse these two challenges include the establishment of protected areas and the restoration of degraded ecosystems [5,6]. Protected areas are considered to be an effective tool for harmonizing the troubled relationship between humanity and the rest of nature [7–9], especially in biodiversity hotspots such as mangrove forests and coral reefs [10,11]. Protected areas also play a critical role in the mitigation of global climate change [12].

To sustainably use and conserve oceans and marine resources, United Nation's Sustainable Development Goals (UNSDG) 14.5 set the target that at least 10% of coastal and marine areas should be protected by 2020 [13]. In 2010, the Convention on Biological Diversity set the 20 "Aichi Targets" for global biodiversity and ecosystem conservation [14]. According to Aichi Target 11, at least 10% of coastal and marine areas should be protected by 2020 [14]. In 2016, Pulitzer Prize winner Edward O. Wilson initiated the Half-Earth project, which aspires to protect 50% of the earth's surface [15]. Additionally, with the increase in the area of degraded ecosystems, there is an urgent need for interventions to restore biodiversity, ecological functioning, and the supply of goods and ecological services [16,17]. Restoration has been widely considered to be an effective way of reversing forest degradation [16,18,19]. For example, the implementation of ecological restoration projects through forestation has significantly increased carbon sequestration by terrestrial ecosystems in China [19].

Mangrove forests are particularly important for biodiversity and ecosystem services. They act as important nursery grounds and breeding sites for ecologically and commercially important animal species, a renewable source of wood, long-term storages for carbon, and a shield against coastal hazards [20]. The value of ecosystem services rendered by mangrove forests is estimated at \$193,845 ha$^{-1}$ yr$^{-1}$, second only to that of coral reefs [21]. The ecological, environmental and socio-economic importance of mangrove forests has been widely recognized and substantial efforts have been invested in their protection and restoration. The rate of mangrove loss shows a decreasing trend, from 1.03% per year in 1980 to 0.67% during the period 2000–2005 [22], and recent analyses have suggested further reduced rates of loss in historical hotspots such as Southeast Asia [23,24]. Despite its relatively small contribution to the global mangrove forest area, China has invested significant resources into mangrove protection and restoration in the last two decades [25]. The mangrove forest area in mainland China decreased from 48,300 ha in 1950 to 22,025 ha in 2001 and China has been listed as the country with the greatest relative losses of mangrove forests of the world [9,26,27], but has been increasing by about 1.8% per year since then. In 2019, the total area of mangrove forests in China was about 30,000 ha [28]. Forty-seven percent of all mangrove forests were protected in 1997 [29]. As of 2019, a total of 43 mangrove protected areas have been established in mainland China, with approximately 67% of the mangrove forest area enclosed in protected areas, far exceeding both the global mean (6.9%) [30] and that of Southeast Asia (20%) [31]. In terms of the proportion of mangrove forests being protected, China has achieved the conservation target of "Nature Needs Half" [8]. Due to strict protection and large-scale restoration, China is one of a very few countries in the world with a recent net increase in mangrove area [23,32].

Vulnerability assessment is the premise for implementing conservation actions to halt and reverse biodiversity loss and ecosystem degradation [33,34]. At present, vulnerability assessments have been made predominantly at the species level [33,35], with only a few conducted at the ecosystem level [34,36]. While some efforts have been made to incorporate multiple-level indicators on mangrove ecosystem health [37], there are yet no reports simultaneously assessing the vulnerability at the species, population, community, and ecosystem levels. Most assessments on mangrove degradation have been focused on the decrease in mangrove forest area [22,38,39], whereas the loss of biological and structural diversity as well as ecosystem services is not well documented. Moreover, mangrove managers are satisfied with the apparent end of area loss, and tend to ignore the loss of ecosystem services caused by alterations in the forest biological and structural diversity [25], which could be counter-productive in mangrove conservation [40].

Mangrove forests are being 'squeezed' by growing coastal populations from the landward fringe and rising sea level from the seaward fringe. The pressures include the synergistic impacts of seawall construction, aquaculture, overfishing, sea level rise, extreme climatic events, ecological invasion, and pollution [41,42]. All these driving factors interact, resulting in the degradation and potential future loss of mangrove forests. The Chinese coastline is prone to tropical typhoon attack. Traditional hard-engineering solutions, such as seawalls, are the preferred option for coastal protection against

storm surges. By 2015, a total of 14,500 km of seawalls had been constructed along the coasts of mainland China, longer than that of the famous ancient "Great Wall" [43].

A few recent reports have suggested that anthropogenic barriers (seawalls) to landward migration precedes all other pervasive anthropogenic disturbances (e.g., aquaculture, pollution, and overfishing) to coastal wetlands [1,44–46]. In China, seawalls have been claimed as the most important pressure to mangrove forests [25,43]. However, there are only a few studies that evaluate the effect of seawalls on coastal wetlands [43].

Degradation of mangrove ecosystems has drawn widespread attention due to their high susceptibility to anthropogenic disturbance and global changes (e.g., rising sea levels) [1,3,47,48]. Degradation must be considered as a major issue in future mangrove conservation and management [49]. There is an urgent need to evaluate the status of degradation with an integrative, multi-level perspective and identify its root cause in order to suggest and implement remedial measures toward timely restoration and management. This paper aims to (1) quantify the degradation of mangrove forests in China by assessing their condition in terms of species diversity, population, and vegetation structure, as well as function, particularly, carbon sequestration; (2) analyze the driving forces behind these; (3) examine the effectiveness of mangrove conservation actions; and (4) propose recommendations for future protection, restoration, and management of mangrove forests in China. This analysis would be valuable to other developing countries facing the same challenges in mangrove conservation.

## 2. Data and Methods

### 2.1. Mangrove Forests in China

The mangrove forests span over seven latitudes from Sanya of the Hainan Province (18°09′ N) to Fuding of the Fujian Province (27°20′ N) in China (Figure 1). The climate type is a tropical/subtropical maritime monsoon climate, with high temperature and rainfall in summer and low temperature and little rainfall in winter. Most mangrove forests are distributed in Hainan, Guangdong and Guangxi, which account for 96% of mangrove forests in China. Fujian, Taiwan, Hongkong and Macau, which also has small areas of mangrove forests. There is no natural distribution of mangrove forests in Zhejiang Province, and all the mangrove forests in Zhejiang Province were introduced in the 1950s. Owing to its north-limit distribution of mangrove forests, the mangrove forests in China are dominated by low-temperature tolerant species such as *Kandelia obovata*, *Aegiceras corniculatum* and *Avicennia marina*. The majority of mangrove forests in China are distributed in estuaries, and characterized by abundant inputs of freshwater and nutrients [27].

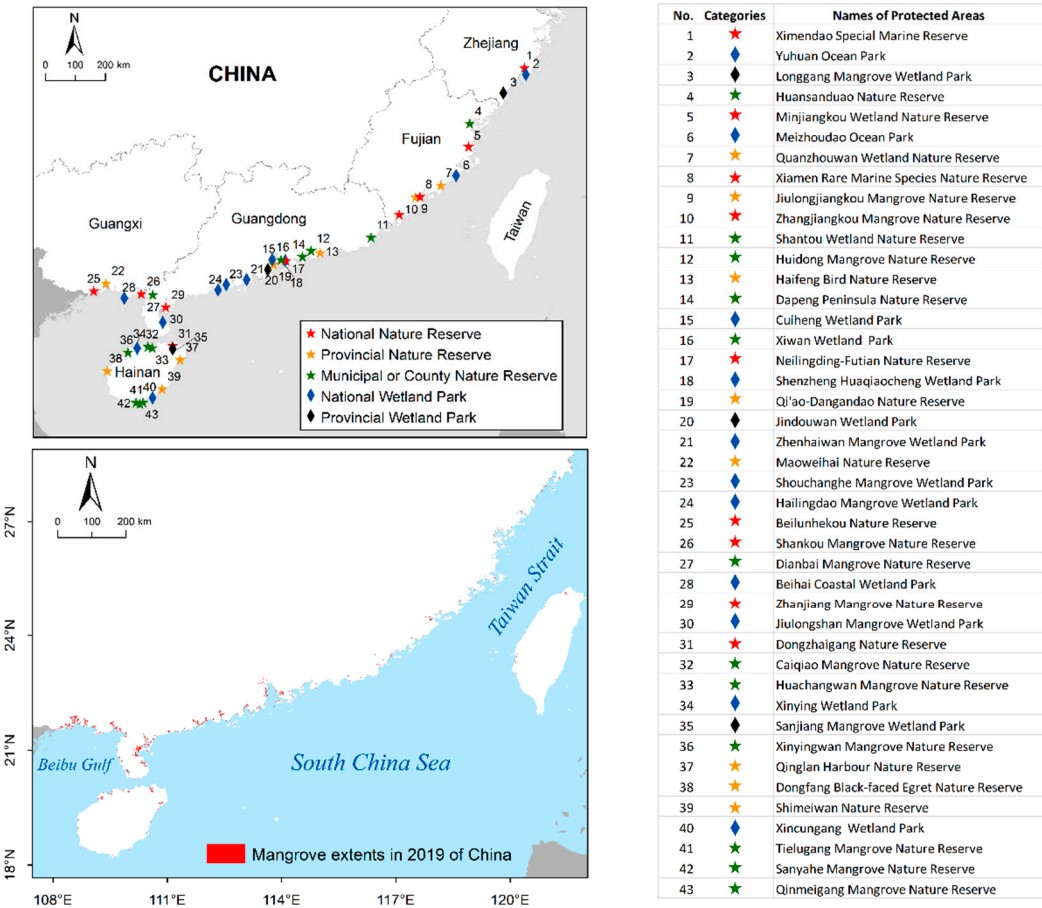

**Figure 1.** Distribution of mangrove forests and the mangrove protected areas in mainland China.

*2.2. Species Diversity and Population Dynamics of Mangrove Communities*

The definition of mangrove species adopted in this study was based on the criteria of Tomlinson [50]. We defined destruction as a change from mangrove to non-mangrove communities, and degradation as the degenerative changes in structure and function of mangrove forests. Data on mangrove species distributions were based on our field investigations from 2009 to 2019 along the southern coastline of China. Red list category was assessed based on the IUCN (International Union for Conservation of Nature) criteria at regional levels in 2012 [51]. All mangrove patches with an area of >1000 m$^2$ were investigated, covering 90% of the mangrove area in China. At each site, mangrove species composition, population size, and community characteristics were recorded. Degradation of mangrove forests in Hong Kong, Macao, and Taiwan were not assessed due to differences in management strategies.

*Bruguiera gymnorhiza* is a dominant mangrove species of China [52], which is widely distributed in the high intertidal zone prone to anthropogenic destruction [50]. The population size of *B. gymnorhiza* was surveyed at 68 sites where this species had been previously reported to explore the impacts of human disturbance on mangrove occurrence. Mature individuals (the tree with height > 1.5 m) were counted at all sites. Populations with n > 300 were merged into one group. Populations with n < 300 were divided into six groups according to population size: 0, 1–10, 11–20, 21–50, 51–100, and 101–300 individuals.

*2.3. Community Structure*

Mangrove forests tend to show sequential species zonation in the intertidal zone: low intertidal forest (i.e., pioneer forest), middle intertidal forest and high intertidal forest on an increasing surface elevation gradient from sea to land [50]. In China, the low-intertidal forest is frequently dominated

by *A. marina* and *A. corniculatum*; the mid-intertidal forest by *K. obovata* and *Rhizophora stylosa*; the high-intertidal forest by *B. gymnorhiza* and *B. sexangula* [27]. The National Mangrove Resource Inventory Report in 2002 was the most comprehensive inventory available and provided data on area, distribution, community types, tree height class, and canopy density of mangrove forests in China in 2001 [26]. This report was used to determine the areas of the three forest types in 2001 in China. In addition, we investigated the community composition of mangrove forests at Dongzhaigang Bay (Hainan, China) in 2014 where mangrove forests suffered comparatively low anthropogenic disturbances [53]. Then, simple comparison of community composition was made between mangrove forests in 2001 and less-disturbed mangrove forests at Dongzhaigang Bay in 2014 to indicate the changes in mangrove community structure.

*2.4. Changes in Carbon Stock and Carbon Sequestration Rate*

The carbon stock and carbon sequestration rate (i.e., net primary productivity in Mg C·ha$^{-1}$ yr$^{-1}$) of native mangrove forests in China were estimated for the years 1950, 2001 and 2019. *A. marina*, *A. corniculatum, R. stylosa, K. obovata*, *B. gymnorhiza*, and *B. sexangula* constituted the overwhelming majority of mangrove forests in China for seven decades (1950–2019) [26–28,52]. The carbon density and net primary productivity (NPP) of each dominant forest type respectively in the low, middle, and high intertidal regions were extracted from the literature, and assumed to be constant between 1950 and 2019. We calculated the carbon density and NPP of low, middle, and high intertidal forests, respectively. Then, temporal variations in carbon stock and carbon sequestration ability of mangrove forests from 1950 to 2019 were estimated based on the changes in areas of these three forest types (i.e., low, middle, and high intertidal forests).

In 1950, the total area of mangrove forests was estimated at about 48,300 ha [27]. There is no published information available on the areas of different types of mangrove forests. The only certainty is that mangrove forests had a less-disturbed community structure in 1950 [52]. Additionally, the mangrove forests at Dongzhaigang Bay (Hainan, China) have been strictly protected since 1980 and had suffered a relatively small amount of anthropogenic disturbance [53]. Therefore, we assumed that the proportions of low intertidal forest, middle intertidal forest and high intertidal forest in 1950 was roughly equal to that at Dongzhaigang. In 2001, the total mangrove forest area in China was estimated at 22,025 ha, and the proportions of low, middle, and high intertidal forests were calculated based on the National Mangrove Resource Inventory Report [26]. In 2019, the total mangrove forest area in China was estimated at 29,227 ha [28]. Data on the areas of three forest types in 2019 were scarce. For simplicity, we assumed that the proportions of low, middle, and high intertidal forests in 2019 were mostly consistent with that in 2001. Then, the carbon stock and carbon sequestration ability of mangrove forests in 1950, 2001, and 2019 were calculated through multiplying the carbon density and net primary productivity (NPP) of each forest type by its corresponding area. Specifically, the proportion of low intertidal forest in 2019 is underestimated because the mangrove forest area increased from 22,025 ha in 2001 to 29,227 ha in 2019 mainly by planting *A. marina, A. corniculatum and K. obovata* (the forests dominated by exotic species not included). Most of the additional mangrove forests are young forests at a low intertidal forest, which have a smaller carbon density and NPP [54,55]. In order to improve the accuracy of estimation, the artificial forests with a coverage less than 30% were also not included. Therefore, our estimations of the vegetation carbon stock and carbon sequestration rate in 2019 are overestimated and the decrease in carbon stock and carbon sequestration rate from 1950 to 2019 may be considered conservative. Table 1 shows the mangrove area and NPP at different times.

**Table 1.** Mangrove areas and net primary productivity (NPP) in 1950, 2001, and 2019 in mainland China.

| | Low Intertidal Forest | Middle Intertidal Forest | High Intertidal Forest | Total |
|---|---|---|---|---|
| Area in 1950 (ha) | 14,973 | 20,286 | 13,041 | 48,300 |
| Area in 2001 (ha) | 11,673 | 8370 | 1982 | 22,025 |
| Area in 2019 (ha) | 13,105 | 9396 | 2226 | 24,727 * |
| NPP in 1950 (Mg·C·yr$^{-1}$) | 118,889 | 183,678 | 178,596 | 481,169 |
| NPP in 2001 (Mg·C·yr$^{-1}$) | 92,686 | 75,745 | 27,146 | 195,577 |
| NPP in 2019 (Mg·C·yr$^{-1}$) | 104,056 | 85,078 | 30,477 | 219,611 |

*: The forests dominated by exotic species (*Sonneratia apetala* and *Laguncularia racemosa*) were not included.

### 2.5. Landward Margins of Mangrove Forests

The types of landward margins of mangrove forests were investigated along the coastal area of South China to explore the possible drivers of mangrove degradation. Based on the possibility of landward migration of mangrove forests facing sea-level rise, the landward topographic types of mangrove forests were divided into five categories: concrete seawall, earthen seawall, road, steep slope, and lowland wetland. Concrete seawalls made of stones and cements were built to protect larger economic facilities and infrastructures, blocking the landward migration of mangrove forests. Earthen seawalls were built in front of smaller farmlands and aquaculture ponds (<50 ha in general). These latter barriers are prone to damage by storm surges and need routine maintenance, and often leave some space for landward migration. Steep slopes are topographically steep landward margins (slope > 10%) and uninhabitable for mangrove seedling establishment. Lowland wetlands are the sites inundated occasionally by peak spring tides and are typically inhabited by mangrove associates or salt marsh plants. Lowland wetlands offer the best potential for mangrove landward migration. The mangrove forests outside seawalls, roads and steep slopes are unlikely to expand or migrate landward due to topographic constraints [43,44]. The perimeter of each type of landward margin parallel to the shoreline was measured by portable GPS (GPSmap 621sc, GARMIN Inc., Taiwan, China).

## 3. Results

### 3.1. Mangrove Species Diversity and Population Dynamics

Based on a long-term (2009–2019) field investigation, the occurrence of 26 species of the mangroves from 12 families was confirmed (Table 2). Of the 26 mangrove species in China, 13 (50%) were categorized as either Critically Endangered (CR, 4 species), Endangered (EN, 4 species) or Vulnerable (VU, 5 species) (Table 2). Severely endangered species with restricted distribution and small population sizes included *Lumnitzera littorea*, *Sonneratia* × *hainanensis*, *S. ovata*, and *Rhizophora* × *lamarckii*. *L. littorea* was only naturally distributed at the Southeastern part of Hainan Island (two small populations). Only one population with 11 individuals was found in our recent survey in June 2019, whereas the population size was 359 in 2006 (Figure 2). *S.* × *hainanensis* was only naturally distributed at Qinglan Bay in Hainan Island. Its population size changed from 40 in 2006 to 29 in 2015. *S. ovata* was only recorded at Qinglan Bay, with a population size of 89. No seedlings of these three species have been recorded in the field. *R.* × *lamarckii* was naturally distributed at Xincun Bay and Xinying Bay of Hainan Island with a population size of 39. Although *B. gymnorhiza* was not classified as endangered according to its overall occurrence in China, extinction or endangered status was common in localized areas. Of the 68 sites where *B. gymnorhiza* was historically distributed, it was extirpated in 45.6% of the sites; 19.1% had a population of less than 50 individuals, and only 11.8% had a population of more than 300 individuals (Figure 3).

**Table 2.** Natural regional distribution of mangrove species in China and their IUCN Red List status. CR: Critically Endangered, EN: Endangered, VU: Vulnerable, NT: Near Threatened, LC: Least Concern, NE: Not Evaluated.

| Family | Species | Intertidal Position [56] | | | Red List Category in China | Global Red List Category [57] |
|---|---|---|---|---|---|---|
| Acanthaceae | *Acanthus ebracteatus* | | M | H | EN | LC |
| | *A. ilicifolius* | | M | H | LC | LC |
| Palmae | *Nypa fruticans* | L | M | H | VU | LC |
| Avicenniaceae | *Avicennia marina* | L | | | LC | LC |
| Combretaceae | *Lumnitzera littorea* | | | M | CR | LC |
| | *L. racemosa* | | M | H | LC | LC |
| Acrostichaceae | *Acrostichum aureum* | | | H | LC | LC |
| | *A. speciosum* | | | H | EN | LC |
| Euphorbiaceae | *Excoecaria agallocha* | | M | H | LC | LC |
| Meliaceae | *Xylocarpus granatum* | | M | H | VU | LC |
| Myrsinaceae | *Aegiceras corniculatum* | L | | | LC | LC |
| Rhizophoraceae | *Bruguiera gymnorhiza* | | M | H | LC | LC |
| | *B. sexangula* | | M | H | NT | LC |
| | *B. s.* var. *rhynchopetala* | | M | H | VU | NE |
| | *Ceriops tagal* | | M | H | LC | LC |
| | *Kandelia obovata* | L | M | | LC | LC |
| | *Rhizophora apiculata* | | M | | VU | LC |
| | *R. stylosa* | L | M | | LC | LC |
| | *R.* × *lamarckii* | | M | H | CR | NE |
| Rubiaceae | *Scyphiphora hydrophyllacea* | | | H | EN | LC |
| Sonneratiaceae | *Sonneratia alba* | L | | | LC | LC |
| | *S. caseolaris* | L | | | NT | LC |
| | *S.* × *gulngai* | | | H | EN | NE |
| | *S.* × *hainanensis* | | | H | CR | NE |
| | *S. ovata* | | | H | CR | NT |
| Lythraceae | *Pemphis acidula* | | | H | EN | LC |

Note: L = low, M = medium, H = high. The red list category worldwide is based on "The IUCN Red List of Threatened Species" (Version 2017-3) [58].

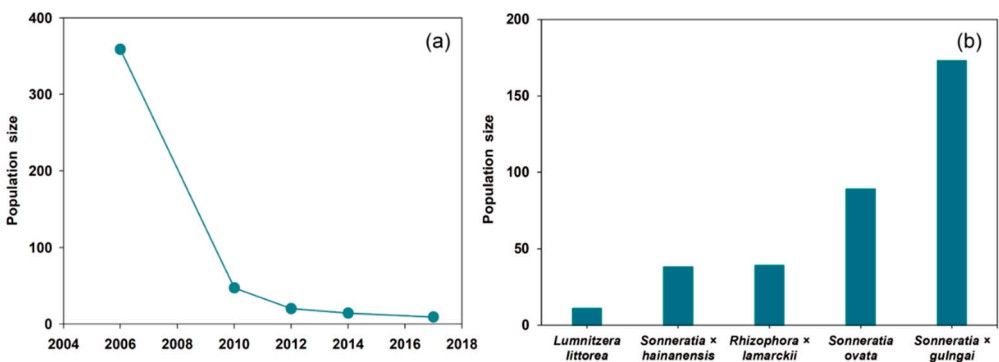

**Figure 2.** Changes in the population size of *Lumnitzera littorea* (**a**) and the population sizes of five of the most endangered mangrove species in China in 2019 (**b**).

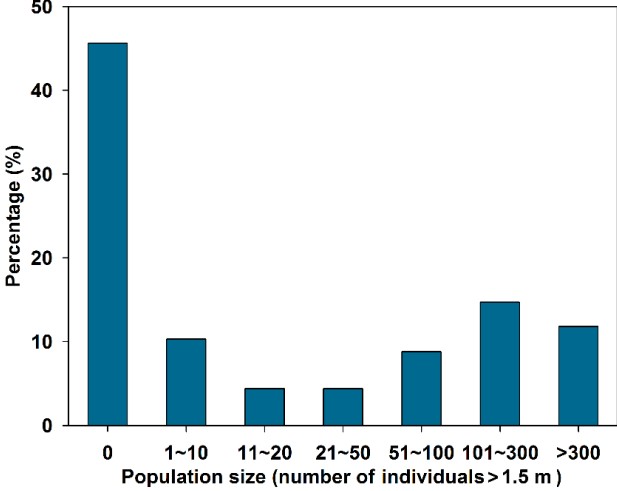

**Figure 3.** Distribution of population size of *B. gymnorhiza* at 68 sites where the species was historically recorded. Population size was divided into seven groups. A population size of 0 indicates that *B. gymnorhiza* has become locally extinct, while a population size > 0 indicates that *B. gymnorhiza* was present during our field investigation (2009–2019).

### 3.2. Simplification of Community Structure

Dongzhaigang Bay (Hainan Island, China) was designated a mangrove reserve in 1980 and a Ramsar Site of International Importance in 1992. It has the highest management level and the least amount of anthropogenic disturbance among the mangrove sites in China. The area of mangrove forest at the bay is 1733 ha. The proportions of low intertidal forest, middle intertidal forest and high intertidal forest at Dongzhaigang Bay in 2014 were 31%, 42% and 27%, respectively (Figure 4). However, when the analysis was enlarged to encompass the whole collection of mangrove forests in China, the relative areas were 53%, 38% and 9%, respectively in 2001 (Figure 4). In comparison to Dongzhaigang Bay where mangrove forests are less disturbed, China has a greater proportion of low intertidal forest and a smaller proportion of high intertidal forest. The majority of mangrove forests in China are the low intertidal, pioneer forest.

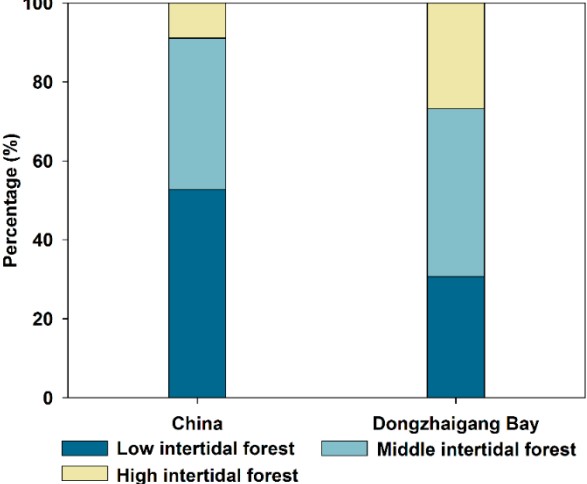

**Figure 4.** Proportions of low intertidal forest, middle intertidal forest, and high intertidal forest in China and Dongzhaigang Bay. Dongzhaigang Bay (Hainan, China) had the highest management level and the least amount of anthropogenic disturbance.

### 3.3. Reduced Carbon Stock and Carbon Sequestration Rate

The vegetation carbon stock of native mangrove forests in China decreased by 58.3% from approximately 21.8 Tg C in 1950 to 9.1 Tg C in 2001, due to a 54.4% loss of mangrove forest area primarily in high-intertidal habitats. From 2001 to 2019, the vegetation carbon stock increased to 10.2 Tg C due to the gains in mangrove forest area. In addition, the carbon sequestration rate (i.e., NPP) of native mangrove forests in China showed a decreasing trend, declining by 10.8% from 10.0 Mg C ha$^{-1}$ yr$^{-1}$ in 1950 to 8.9 Mg C ha$^{-1}$ yr$^{-1}$ in 2001 and 2019 (Figure 5).

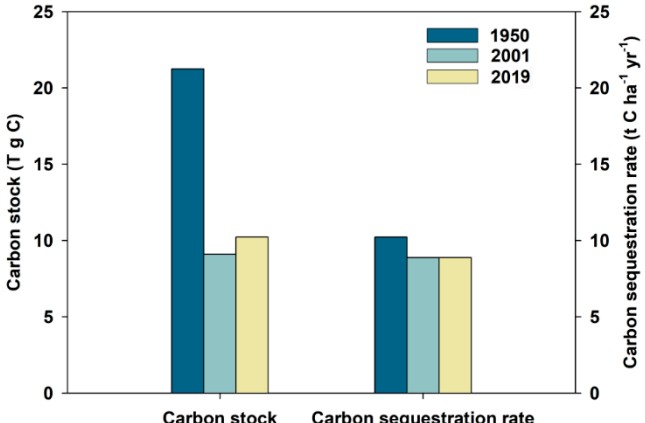

**Figure 5.** Changes in carbon stock and carbon sequestration rate (i.e., net primary productivity) of native mangrove forests in China.

### 3.4. Breakdown of Landward Forest Margins

There were four types of mangrove landward margins in China (i.e., concrete seawall, earthen seawall, road, and steep slope). The percentages of the four landward margin types were as follows: steep slope (3.0%), road (4.0%), earthen seawall (10.0%), and concrete seawall (83.0%) (Figure 6). Seawall was by far the dominant type of mangrove landward margins in China. Blocked by the seawall, there is no natural landward habitats (i.e., lowland wetland), which are suitable for mangrove landward migration.

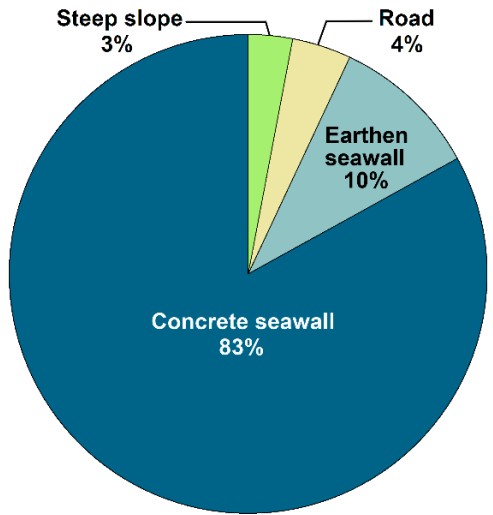

**Figure 6.** Composition of landward topographic types of mangrove forests in China.

## 4. Discussion

### 4.1. Rampant Degradation Despite Strict Protection and Increase in Mangrove Forest Area

Globally, most mangrove species are increasingly threatened and are undergoing contraction due to local extinction [59–61]. Although no mangrove species in China have been included on the list of globally threatened species [57] and there is no record of the extinction of mangrove species as yet (Table 2), the distribution ranges of some species and related wild resources have been drastically reduced. Of the 26 mangrove species in China, 13 (50%) were assessed as CR, EN or VU. In contrast, of the 35,784 higher plant species in China, only 10.8% were assessed as CR, EN or VU [62]. Moreover, mangrove species in China showed a higher probability of extinction than the global estimate (16%), even higher than that (40%) in the Atlantic and Pacific coasts of Central America [57]. Local extinction of species such as *B. gymnorhiza* was common in China (Figure 3). Therefore, the high percentages of locally extinct or endangered species were the most remarkable indictors of mangrove degradation in China.

Degradation may be reflected by changes in forest structure [63], such as small fragmented mangrove communities [49], reduced density of robust trees and extinction of some species [64]. In China, the four species of Rhizophoraceae (*B. gymnorhiza*, *B. sexangula*, *K. obovata*, and *R. stylosa*) dominated the mangrove forests sixty years ago [52], while the most dominant species in 2001 were *A. corniculatum*, *A. marina* and *K. obovata* characteristic of the low intertidal forests [26]. Fifty-three percent of mangrove forests in China were low intertidal forests, and only 9% were high intertidal forests. This means that loss from 1950 to 2001 mostly occurred in the high intertidal forests. There was a dramatic shift in the mangrove forest structure along with the decreased forest area. In addition, the changes in mangrove forest structure were consistent with the loss of mangrove species. The species assessed as CR, EN or VU are all distributed at the high or middle intertidal zones (Table 2). Local extinction and range contraction of endangered mangrove species such as *Heritiera fomes* and abnormal expansion of native species such as *Ceriops decandra* have also been reported in Sundarbans [61]. These results also suggest what are being protected are not the diverse, healthy forests that used to occur. Low species diversity and low structural complexity are characteristic of degraded mangrove forests in mainland China.

Mangrove forests in China also showed typical properties of ecosystem retrogression characterized by a reduction in their ecosystem productivity and carbon sequestration rates [65]. In Southeast Madagascar, the degradation of mangrove forests resulted in a 25.1% decrease in carbon stock (from $454.9\,\mathrm{Mg\,C\,ha^{-1}}$ to $340.9\,\mathrm{Mg\,C\,ha^{-1}}$) [63]. In Cambodia, compared to great loss (60%) of carbon stock, the degradation from sustainable harvesting of small mangrove trees resulted in almost no loss of carbon stock [66]. In China, the carbon stock of native mangrove forests decreased by 53.1%, from 21.8 Tg C in 1950 to 10.2 Tg C in 2019 (Figure 5); meanwhile, the area of native mangrove forests only decreased by 48.8%. This discrepancy was caused by the structural changes of mangrove forests. The carbon density typically increases along the intertidal profile from sea to land, with high intertidal forests having the highest carbon density [54,55]. The destruction of high intertidal forests would lead to greater carbon loss. In addition, a 10.8% decrease in the capacity for carbon sequestration was expected due to the alteration of mangrove forest composition (i.e., replacement of high sequestration communities by low sequestration communities) (Figure 5), leading to a reduction in carbon sequestration by mangrove vegetation of up to 0.26 Tg C·yr$^{-1}$ between 1950 and 2019 at the national scale. Young mangrove forests generally have higher NPP than mature forests [54,55]. However, global climate and anthropogenic disturbances negatively affect NPP [3,18,41,67]. The combination of these two factors result in the no net changes in NPP from 2001to 2019 (Figure 5).

### 4.2. Root Cause of Mangrove Degradation—Seawall Construction

Seawall construction and concomitant aquaculture activities have had a particularly disastrous impact on the Chinese mangrove ecosystems. These anthropogenic activities not only directly destroy

large areas of mangrove forests but also continuously threaten the remaining mangrove forests seaward of the seawalls [25,43]. Mangrove forests typically have a high species diversity, tall trees as well as large biomass and carbon stock in the high intertidal zones [27,50]. However, this zone is almost invariably the first to be reclaimed, and thus most prone to anthropogenic destruction. In order to increase the area of reclaimed lands, most seawalls in China were built in the middle intertidal or even the low intertidal position [27], leading to severe "coastal squeeze" in elevation-deficient scenarios [68,69]. Considering the list of endangered mangrove species, almost all species are distributed in the high intertidal zone (Table 2). Hence, the loss of habitat resulting from seawall construction and wetland reclamation is the major reason for the higher ratio of endangered mangrove species in China. This is also why mangrove forests in China are currently dominated by the low intertidal species (Figure 4). In addition, seawalls destroy sea-to-land connectivity and impact mangrove forests by disrupting hydrodynamic patterns, organism movements, nutrient flows and modifying sediment balance [1,43–45]. Mangrove forests constrained on their landward side by a seawall often contain less leaf litter and have fewer saplings than forests without seawalls [70]. Moreover, aquaculture (shrimp or fish farming) can reduce mangrove growth rates, increase mangrove mortality rates, and lead to ecological degradation through excess pond waste materials in sediments discharged from nearby aquaculture ponds [67,71].

### 4.3. The Effectiveness of Mangrove Conservation Actions

Despite the increase in mangrove forest area, there has been an overall gradual simplification of mangrove forests in China. Mangrove forests in China are suffering extensive degradation resulting from biological and structural changes. Neither the expanded coverage of protected areas nor the current restoration have been sufficient to conserve the biological and structural diversity as well as the related ecological services of these forests. The compromised effectiveness of mangrove conservation efforts is related to the following existing strategies and approaches in mangrove conservation.

First, mangrove managers have been concerned exclusively about forest area. In contrast, some key environmental (e.g., water quality, hydrological regime) and biological (e.g., multi-taxa biodiversity) components of the mangrove ecosystem receive relatively little attention from managers. Mangrove forests connect naturally with adjoining ecosystems such as the seaward seagrass beds and landward freshwater swamps, which is fundamental to mangrove ecosystem function and services [72,73]. However, most natural landward margins of mangrove forests in China have been destroyed by seawall construction (Figure 6). Second, current mangrove restoration projects focus on the plantation on bare mudflats [25,74]. Most of the restoration attempts were proven to be unsuccessful due to the severity of inundation stress on bare mudflats [74]. In addition, this kind of restoration would decrease species richness and abundance of water birds because the mudflat area outside mangrove forests is the major foraging site for migratory waterbirds [9,27,75], as well as having more frequent inundation. Third, mangrove restoration projects have so far focused on a low diversity of species to satisfy forest production and economic interests. Specifically, fast-growing exotic mangrove species (*Sonneratia apetala*, *Laguncularia racemosa*) have been extensively used in many mangrove restoration projects in the last two decades, accounting for 62% of the increased forest areas (2001–2019). These exotic mangrove species display high adaptability and growth rate, with an average height greater than forests comprising native species [76]. While large-scale restoration using exotic species like *S. apetala* could significantly increase the mangrove carbon stock, this practice reduces the biodiversity of the replanted forests, suppresses the development of native mangrove forests, and may lead to increased risks of biological invasion [76,77].

### 4.4. Conservation Priorities and Perspectives

Due to the high population density in the coastal areas of South China (700 individuals per km$^2$) [78], there is no possibility of conducting large-scale managed retreats by breaching old seawalls [79]. The adverse effects of seawalls and aquaculture (shrimp or fish farming) on the adjacent mangrove forest will persist over long periods of time. However, some remedial measures can be taken to

hopefully halt and reverse mangrove degradation in China. Our result showed that about 10% of the landward margins of mangrove forests were backed by aquaculture ponds or farmlands with earthen seawalls (Figure 6). These seawalls are easily destroyed by storm surges and need frequent repair, which leads to increased costs of farming; and consequently, a number of aquaculture ponds were permanently abandoned. This may create room for mangrove landward migration or opportunities for replanting in abandoned ponds [9,25,31]. Therefore, with careful planning, some abandoned ponds in certain regions can be restored to mangrove wetlands by breaching old earthen seawalls. Similarly, some abandoned ponds inside concrete seawalls can be restored to mangrove wetlands by judiciously restoring the tidal connectivity [44]. These solutions will not only enlarge the area of mangrove forests, but also have the potential to increase the resilience of coastal wetlands to disturbances caused by global climate change [80]. Therefore, we recommend that the Chinese government halt mangrove planting on bare mudflats and where possible, prioritize the conversion of aquaculture ponds into mangrove wetlands to increase forest area and resilience to disturbances. In addition, the structure, function, and ecological processes in mangrove ecosystems should be integrated into the goals and deliverables of mangrove restoration projects. Restoration projects should be carried out by mainly planting more native mangrove species; and introduction of fast-growing exotic mangrove species to dwindling native forests should be forbidden. A shift in management strategy from an area-based to an ecosystem function-based strategy is urgentlys needed to ensure the long-term sustainability of mangrove resources in China.

## 5. Conclusions

This paper provides the baseline data for the proper conservation, management and restoration of mangrove resources in China. With the implementation of the Environmental Protection Law of the People's Republic of China in 1999 and other laws and regulations, the direct destruction of mangrove forests has been legally prohibited since 2001, and the area of mangrove forests in China has shown signs of improvement due to strict protection of the remaining mangrove forests and large-scale restoration. However, indirect and subtle threats resulting from widespread coastal wetland reclamation through seawall construction have serious negative implications for the long-term sustainability of mangrove forests at both the species and community levels. These threats can also compromise the functional integrity of the ecosystems and their capacity to provide ecosystem services. Without changes in protection and recovery strategies, mangrove forests will continue to be degraded, with lower species richness, structural complexity, carbon sequestration and storage capacity, and resilience to disturbances. Current mangrove restoration implemented mainly by planting exotic, fast-growing species on bare mudflats will lead to irreversible loss of biodiversity and related ecological services. With the intensification of climate change and anthropogenic disturbances, there is an urgent need for science-based mangrove coastal management to halt, and hopefully reverse, the rampant degradation of mangrove forests in China.

**Author Contributions:** W.W. designed the experiments and wrote the manuscript. H.F. (Haifeng Fu) collected and analyzed part of the data. S.Y.L. and H.F. (Hangqing Fan) reviewed and edited the manuscript. M.W. conceived the ideas. All authors have read and agreed to the published version of the manuscript.

**Funding:** The work was jointly supported by the National Natural Science Foundation of China (31670490, 41276076); and the Programs of Science and Technology on Basic Resources Survey for the Ministry of Science and Technology of China (2017FY100701).

**Acknowledgments:** We would like to thank LetPub (www.letpub.com) for providing linguistic assistance during the preparation of this manuscript.

**Conflicts of Interest:** The authors declare no conflicts of interest.

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
