# Peer review of "Can Strict Protection Stop the Decline of Mangrove Ecosystems in China? From Rapid Destruction to Rampant Degradation"

_forests, doi:10.3390/f11010055_

Round 1

Reviewer 1 Report

This is a clearly written timely article with practical implications for China’s mangrove management and conservation. Objectives are specific, methods are robust, results are transparent and the conclusions are valid. I enjoyed reading this article. Thanks to the authors. However, I think a little bit of work (mentioned below) is required to make the article more interesting to the readers.

In the ‘Data and Methods’ section inclusion of a map showing the distribution of the mangrove forests (also the mangrove Protected Areas) in China would help the readers. In addition, a brief description about the forests in terms of vegetation, soil, climate and hydrological variability would help the readers to understand the China’s mangrove systems better. This would also make the messages of this paper clearer to the wider audience.

Referring recent works on other mangrove ecosystems facing similar problems may connect the outcomes of the paper to broader community. For example, the statement in Line 268: ‘Globally, most mangrove species are increasingly threatened and are undergoing contraction due to local extinction’ - may be supported by two recent relevant papers:

Sarker, S. K., J. Matthiopoulos, S. N. Mitchell, Z. U. Ahmed, M. B. Al Mamun, and R. Reeve. 2019. 1980s–2010s: The world’s largest mangrove ecosystem is becoming homogenous. Biological Conservation 236:79–91.

Sarker, S. K., R. Reeve, N. K. Paul, and J. Matthiopoulos. 2019. Modelling spatial biodiversity in the world’s largest mangrove ecosystem-The Bangladesh Sundarbans: A baseline for conservation. Diversity and Distributions 25:1–14.

Author Response

Reviewer 1:

This is a clearly written timely article with practical implications for China’s mangrove management and conservation. Objectives are specific, methods are robust, results are transparent and the conclusions are valid. I enjoyed reading this article. Thanks to the authors. However, I think a little bit of work (mentioned below) is required to make the article more interesting to the readers.

In the ‘Data and Methods’ section inclusion of a map showing the distribution of the mangrove forests (also the mangrove Protected Areas) in China would help the readers. In addition, a brief description about the forests in terms of vegetation, soil, climate and hydrological variability would help the readers to understand the China’s mangrove systems better. This would also make the messages of this paper clearer to the wider audience.

Accepted and revised accordingly. See Figure 1 and Part 2.1.

Referring recent works on other mangrove ecosystems facing similar problems may connect the outcomes of the paper to broader community. For example, the statement in Line 268: ‘Globally, most mangrove species are increasingly threatened and are undergoing contraction due to local extinction’ - may be supported by two recent relevant papers:

Sarker, S. K., J. Matthiopoulos, S. N. Mitchell, Z. U. Ahmed, M. B. Al Mamun, and R. Reeve. 2019a. 1980s–2010s: The world’s largest mangrove ecosystem is becoming homogenous. Biological Conservation 236:79–91.

Sarker, S. K., R. Reeve, N. K. Paul, and J. Matthiopoulos. 2019b. Modelling spatial biodiversity in the world’s largest mangrove ecosystem-The Bangladesh Sundarbans: A baseline for conservation. Diversity and Distributions 25:1–14.

Accepted and revised accordingly. We added the two references.

Reviewer 2 Report

I enjoyed this short paper that analyzes changes in the extent and composition of mangrove forests in China.  After decades of loss of mangroves - at least in aerial extension - mangrove ecosystems are making some recuperation - at least in area.  Neverthess, the species composition is changing as is the location of mangroves.  Restoration is using exotic fast-growing mangrove species, and there has been an overall loss of this ecosystem in the high intertidal zone.  The manuscript is well-written, scientifically sound, and concise.

Line 24 - Please confirm that authors are referring to the carbon stock of mangroves.  Does this only include the vegetation biomass or also the carbon stored in sediments?

Line 26 - root cause for this degradation in China is seawall construction

Line 40 - Protected areas also play a critical role .  .  .

Line 43 - Authors are correct to mention Aichi Target 11.  However, I think that they should also mention the UNSDG Target 14.5 (10% marine and coastal areas protected by 2020).

Line 48 - widely considered to be  an effective way .  .  .

Lines 60-61 - China has invested significant resources in mangrove protection

Lines 64-67 - So today 67% of mangroves in China are included in PAs.  I note that PAs are not the only way to protect mangrove ecosystems.  It is possible to prohibit the conversion of mangroves or requiring a policy of "no net loss of mangroves or wetlands" without formally declaring PAs.  However, perhaps PAs (if enforced) are the clearest way to protect this important ecosystem.

Line 79 - with the apparent end of area loss

Line 87 - solutions, sich as seawalls, are the preferred

Line 93 - there are few studies that evaluate the effect of seawalls

Line 102 - vegetation structure, as well as funciton,

Lines 135-140 - If I understand correctly, this seems to be quite an assumption - the comparision of relatively pristine mangrove forests in Dongzhaigang Bay in Hainan with mangrove forests in the rest of the country. Essentially, the Hainan forests serve as a control or baseline for comparative purposes.  Authors should be clearer about this and also provide more justification.  This assumption also appears in Lines 157-159.

Line 149 - intertidal forest, respectively.

Lines 170-174 - Authors should clarify this a little.  Planted mangroves have low biomass initially.  Reforesting 1 ha of mangroves now is extremely different in biomass, carbon sequestration, and ecosystem services than a mature mangrove forest.  Is this what the autors refer to in Lines 173-174?

Lines 194-195 - occurrence of 26 species of the mangroves from 12 families was confirmed

Line 207 - "at a smaller scale" .  .  .  Perhaps "in locaized areas" would be better.

Figure 2 - The horizontal axis needs a date.

Lines 251-252 - Can this really be the case???? Carbon stock increased from 2011 to 2019?????  Many of the afforested areas have extremely low initial biomass.

Line 262 - "There are no natural landward habitats .  .  .  Really???? I am sure that there must be some lowland wetlands landward of mangrove forests somewhere in China.

Line 329 - extensive degradation resulting from biological .  .  .

Line 331 - of these forests.

Line 363 - to increased costs of farming;

Line 364 permanently abandoned.

Line 371 - that the Chinese government halt mangrove afforestation .  . 

Line 375 - Afforestation projects should be carried out .  .  .

Line 381-382 - Direct destruction of mangrove forests has bee legally prohibited since 2001 - The authors should cite a law or regulation to support this claim.

Line 384 - widespread coastal wetland reclamation .  .  .

Line 387 - their capacity to provide ecosystem services.

I am attaching a PDF of a relevant chapter in COASTAL WETLANDS that might be of use to the authors.

Author Response

Reviewer 2:

I enjoyed this short paper that analyzes changes in the extent and composition of mangrove forests in China.  After decades of loss of mangroves - at least in aerial extension - mangrove ecosystems are making some recuperation - at least in area. Nevertheless, the species composition is changing as is the location of mangroves. Restoration is using exotic fast-growing mangrove species, and there has been an overall loss of this ecosystem in the high intertidal zone. The manuscript is well-written, scientifically sound, and concise.

Line 24 - Please confirm that authors are referring to the carbon stock of mangroves.  Does this only include the vegetation biomass or also the carbon stored in sediments?

Yes, this only includes the carbon stored in mangrove vegetation. To avoid misunderstanding, we clarified it.

Line 26 - root cause for this degradation in China is seawall construction

Accepted and revised accordingly.

Line 40 - Protected areas also play a critical role .  .  .

Accepted and revised accordingly.

Line 43 - Authors are correct to mention Aichi Target 11.  However, I think that they should also mention the UNSDG Target 14.5 (10% marine and coastal areas protected by 2020).

Accepted and revised accordingly.

Line 48 - widely considered to be an effective way .  .  .

Accepted and revised accordingly.

Lines 60-61 - China has invested significant resources in mangrove protection

Accepted and revised accordingly.

Lines 64-67 - So today 67% of mangroves in China are included in PAs. I note that PAs are not the only way to protect mangrove ecosystems.  It is possible to prohibit the conversion of mangroves or requiring a policy of "no net loss of mangroves or wetlands" without formally declaring PAs. However, perhaps PAs (if enforced) are the clearest way to protect this important ecosystem.

Yes, we agree with you. China has lost 57% of its coastal wetlands since 1950s. “No net loss of wetlands” is very important. However, we should not satisfied with “no net loss of wetlands”. Wetland type, quality and function should be taken as important indexes of wetland protection. The Chinese government has taken note of the situation and is taking a series of measures.

Line 79 - with the apparent end of area loss

Accepted and revised accordingly.

Line 87 - solutions, such as seawalls, are the preferred

Accepted and revised accordingly.

Line 93 - there are few studies that evaluate the effect of seawalls

Accepted and revised accordingly.

Line 102 - vegetation structure, as well as function

Accepted and revised accordingly.

Lines 135-140 - If I understand correctly, this seems to be quite an assumption - the comparison of relatively pristine mangrove forests in Dongzhaigang Bay in Hainan with mangrove forests in the rest of the country. Essentially, the Hainan forests serve as a control or baseline for comparative purposes. Authors should be clearer about this and also provide more justification. This assumption also appears in Lines 157-159.

Accepted and revised accordingly. According to a recent assessment by Wang et al. (2011), compared to the mangrove forests in the rest of China, Hainan has more pristine mangrove forests. Of all the mangrove forests in Hainan, the mangrove forests of Dongzhaigang Bay are more pristine.

Line 149 - intertidal forest, respectively.

Accepted and revised accordingly.

Lines 170-174 - Authors should clarify this a little.  Planted mangroves have low biomass initially.  Reforesting 1 ha of mangroves now is extremely different in biomass, carbon sequestration, and ecosystem services than a mature mangrove forest.  Is this what the authors refer to in Lines 173-174?

Accepted and revised accordingly. We believe that artificial mangroves have low biomass initially. With the increase of forest age, both vegetation carbon stock and soil carbon stock increase. There have quite a few reports (Wang et al., 2013; Liu et al., 2014; Alongi & Mukhopadhyay, 2015). In order to improve the accuracy of estimation, the artificial forests with a coverage less than 30% were not included.

Alongi DM, Mukhopadhyay S K. 2015. Contribution of mangroves to coastal carbon cycling in low latitude seas [J]. Agricultural and Forest Meteorology, 213(15): 266-272.

Lines 194-195 - occurrence of 26 species of the mangroves from 12 families was confirmed

Accepted and revised accordingly.

Line 207 - "at a smaller scale" .  .  .  Perhaps "in locaized areas" would be better.

Accepted and revised accordingly.

Figure 2 - The horizontal axis needs a date.

Not accepted. We think the figure has clearly shown the population size grade of Bruguiera gymnorhiza.

Lines 251-252 - Can this really be the case???? Carbon stock increased from 2001 to 2019?????  Many of the afforested areas have extremely low initial biomass.

Yes, we believe that many artificial forests have low initial biomass and low carbon sequestration ability. With the increase of forests age, biomass and carbon sequestration ability increase gradually (Wang et al., 2013; Liu et al., 2014; Alongi & Mukhopadhyay, 2015). From 2001 to 2019, mangrove forest area increased by 32.70% (from 22025 ha to 24227 ha), however the vegetation carbon stock only increased by 12.09 % (from 9.1Tg to 10.2 Tg). The main reason for this is the low initial biomass of artificial forests. To avoid misunderstanding, we revised accordingly.

Line 262 - "There are no natural landward habitats .  .  .  Really???? I am sure that there must be some lowland wetlands landward of mangrove forests somewhere in China.

Yes, there must be some lowland wetlands landward of mangrove forests, including widespread aquaculture ponds. What we meant is that there are no lowland wetlands for mangrove forest landward migration. We have clarified it.

Line 329 - extensive degradation resulting from biological .  .  .

Accepted and revised accordingly.

Line 331 - of these forests.

Accepted and revised accordingly.

Line 363 - to increased costs of farming;

Accepted and revised accordingly.

Line 364 permanently abandoned.

Accepted and revised accordingly.

Line 371 - that the Chinese government halt mangrove afforestation .  . 

Accepted and revised accordingly.

Line 375 - Afforestation projects should be carried out .  .  .

Accepted and revised accordingly.

Line 381-382 - Direct destruction of mangrove forests has been legally prohibited since 2001 - The authors should cite a law or regulation to support this claim.

Accepted and revised accordingly. In the late 1990s, the Chinese government began to pay attention to mangrove protection. A series of laws and regulations have provisions to protect mangroves, including Marine Environmental Protection Law of the PeoplesRepublic of China (1999), Regulations of Guangdong Province on Wetland Protection (2006) and Regulations of Fujian Province on Wetland Protection (2017). Additionally, there are also some local regulations especially for mangrove protection, like Regulations on Mangrove Protection of Hainan Province (1998) and Regulations of Guangxi Zhuang Autonomous Region on the Protection of Mangrove Resources (2018).

Line 384 - widespread coastal wetland reclamation .  .  .

Accepted and revised accordingly.

Line 387 - their capacity to provide ecosystem services.

 Accepted and revised accordingly.

I am attaching a PDF of a relevant chapter in COASTAL WETLANDS that might be of use to the authors.

 Accepted and revised accordingly. We added the reference.

Reviewer 3 Report

Authors have gathered lot of data from literature and also from the field to clarify importance and conservation of mangroves in China and cause of destruction. Overall manuscripts is interesting and readers can gain much more knowledge from this article. This article would be valuable to other developing countries facing the same challenges in mangrove conservation.

Line 153-174: Carbon stock and NPP calculation is quite redundant, as many assumption have been made to estimate CS and NPP. However, it will be useful for researchers to see some numbers and kind of data needed for future research to estimate more accurate CS and NPP.    

Below are my minor comments to improve the manuscript.

Abstract

Line 15: large scale afforestation? is this correct because afforestation means mangroves were planted where in the no mangroves were present. Or it includes restoration and nature regeneration also. More clarification would be beneficial for the readers.   

Keywords: delete keywords which are already in manuscript title and more relevant keywords related to study area can be added.

Introduction 

Line 49: defining restoration vs afforestation is necessary to understand difference between these two terms.

Data and methods

Line 123-124: Need strong justification for choosing more than 1.5 m trees only for the analyses. It depends on the objective however authors have studied community structure and species diversity and sapling and seedling are very important in terms of quantifying mangrove species diversity.  

Results

Line 249-254: Authors have explained why CS values differ in each year, however it would be good to also explain why NPP values in 2001 and 2019 are same. Add explanation 

Discussion

Line 293-295: Many researchers have look into how deforestation leads to loss of mangrove carbon stock. Few have checked degradation and deforestation separatly. Recent papers have explained about degradation and deforestation, would be good to refer them. , The impacts of degradation, deforestation and restoration on mangrove ecosystem carbon stocks across CambodiaScience of The Total Environment10.1016/j.scitotenv.2019.135416(135416)(2019).  

Line 299-304: Any data to show carbon sequestration in soil via sedimentation or sediment C burial rate? It would be good to have that kind of information if we doing long term studies which involves both literature as well as field based data. 

Author Response

Reviewer 3:

Authors have gathered lot of data from literature and also from the field to clarify importance and conservation of mangroves in China and cause of destruction. Overall manuscripts is interesting and readers can gain much more knowledge from this article. This article would be valuable to other developing countries facing the same challenges in mangrove conservation.

Line 153-174: Carbon stock and NPP calculation is quite redundant, as many assumption have been made to estimate CS and NPP. However, it will be useful for researchers to see some numbers and kind of data needed for future research to estimate more accurate CS and NPP.

Accepted. We attached the relative data in Table 2.

Line 15: large scale afforestation? Is this correct because afforestation means mangroves were planted where in the no mangroves were present. Or it includes restoration and nature regeneration also. More clarification would be beneficial for the readers.

Accepted and revised accordingly.

Keywords: delete keywords which are already in manuscript title and more relevant keywords related to study area can be added.

Accepted and revised accordingly.

Line 49: defining restoration vs afforestation is necessary to understand difference between these two terms.

Accepted and revised accordingly.

In China, large areas of mangrove forests have been restored by artificial planting or natural settlement of mangrove seedlings in the stands with mangrove forests in history. We think this is restoration. Additionally, there are some examples that planting mangrove seedlings in the mudflats outside the current mangrove forests. Here we consider it afforestation. Because of low surface elevation, many of these projects failed.

Line 123-124: Need strong justification for choosing more than 1.5 m trees only for the analyses. It depends on the objective however authors have studied community structure and species diversity and sapling and seedling are very important in terms of quantifying mangrove species diversity.  

Accepted and revised accordingly. The numbers of recruited diaspore, density of seedling and sapling are important in terms of evaluating population existence. However, here we just try to determine the population size of Bruguiera gymnorhiza. In general, only mature individual is included. According to our field survey, most Bruguiera gymnorhiza individuals don't start blooming until they are above 1.5 m. To avoid misunderstanding, we revised accordingly.

Line 249-254: Authors have explained why CS values differ in each year, however it would be good to also explain why NPP values in 2001 and 2019 are same. Add explanation.

Accepted and revised accordingly.

Line 293-295: Many researchers have look into how deforestation leads to loss of mangrove carbon stock. Few have checked degradation and deforestation separately. Recent papers have explained about degradation and deforestation, would be good to refer them. Sahadev Sharma, Richard A. MacKenzie, Thida Tieng, Kim Soben, Natcha Tulyasuwan, Amomwan Resanond, Geoffrey Blate and Creighton M. Litton, The impacts of degradation, deforestation and restoration on mangrove ecosystem carbon stocks across Cambodia, Science of The Total Environment, 10.1016/j.scitotenv.2019.135416, (135416), (2019).  

Accepted and revised accordingly.

Line 299-304: Any data to show carbon sequestration in soil via sedimentation or sediment C burial rate? It would be good to have that kind of information if we doing long term studies which involves both literature as well as field based data.

Not accepted. In this paper we only consider the vegetation carbon stock. It would be very good to include the information of soil carbon stock. We tried to estimate the changes in soil carbon stock. However, we couldn’t finish it because of large data gap. Some researchers in our University have noticed this and are collecting needed data.